# Post-Return Stroke VHF Electromagnetic Activity in North-Western Mediterranean Cloud-to-Ground Lightning Flashes

Andrea Kolinska[1,2], Ivana Kolmasova[2,1], Eric Defer[3], Ondrej Santolik[2,1], Stéphane Pédeboy[4]

[1]Faculty of Mathematics and Physics, Charles University, Prague, 121 16, Czechia
[2]Institute of Atmospheric Physics, Czech Academy of Sciences, Prague, 141 00, Czechia
[3]Laboratoire d'Aérologie, Université de Toulouse, CNRS, OMP, UPS, Toulouse, 314 00, France
[4]Météorage, Pau, 640 00, France

*Correspondence to*: Andrea Kolinska (ak@ufa.cas.cz)

**Abstract.** We investigate properties of the electromagnetic activity following the first lightning return stroke (RS), using concurrent observations from the SLAVIA (Shielded Loop Antenna with a Versatile Integrated Amplifier) sensor, the lightning mapping array (LMA) SAETTA (Suivi de l'Activité Electrique Tridimensionnelle Totale de l'Atmosphère) and Météorage LF network in the Corsica region. From the data collected between September and December 2015, we selected 66 negative cloud-to-ground (-CG) and 26 positive cloud-to-ground (+CG) lightning flashes in the north-western Mediterranean region. In the
SAETTA data, we observe a decrease of the Very High Frequency (VHF) radiation rate and the VHF power as recorded within a typical 80-µs time window at the LMA stations, immediately after the RS pulse in 59 -CG flashes. Contrastingly, we show that all examined +CG flashes exhibit a rapid increase of the VHF radiation rate and the VHF power immediately after the RS. We suggest a possible explanation of this phenomenon by considering step-like propagation of a negative part of bidirectional leader starting at the top end of the positive lightning channel inside the thundercloud, emitting electromagnetic radiation
across a broad frequency spectrum.

## 1 Introduction

During the development of lightning flashes, electromagnetic signals are emitted across a wide spectrum of frequencies, detectable by ground-based electric and magnetic antennas. The initial stage of cloud-to-ground (CG) flashes is called a preliminary breakdown (PB) stage (e.g. Clarence and Malan, 1957; Marshall et al., 2014; Karunarathne et al. 2020). Within
this stage, numerous discharges are initiated in the thundercloud and one of them evolves into a bidirectional leader (e.g. Montanya et al., 2015). In broadband electromagnetic recordings (units of kHz to tens of MHz) the PB stage is identifiable as a sequence of bipolar pulses, typically with the same polarity as the upcoming return stroke (RS) (Rakov and Uman, 2003; Kolmasova et al., 2014). However, a study of Wu et al. (2018) showed few cases of positive (+CG) flashes starting with PB pulses with opposite polarity. In the case of negative cloud-to-ground (-CG) flashes, the negative part of the bidirectional
leader channel extends toward the ground, ionizing the air and forming a conductive path for the upcoming return stroke. This

specific phase is referred to as the stepped leader stage where the lightning channel expands in a stepped manner and thus generates intense very high frequency (VHF) radiation (Rakov, 2003). In case of +CG flashes, the downward moving segment is mostly the positive one, while the negative part is moving in the opposite direction (Li et al., 2020; Wu et al., 2018). Positive leaders usually do not radiate in the VHF band at all or not as strongly as the negative leaders (Rakov and Uman, 2003; Shao

et al., 1999).

As the ionized channel of the leader gets closer to the ground, an upward discharge may be initiated and upon its connection with the downward leader, a conductive path between the thundercloud and the ground is established, and the first return stroke (RS) occurs. In case of -CG flashes, negative charge from the cloud begins to move towards the ground in the newly formed

channel, leading to the electric current reaching magnitudes of up to hundreds of kiloamperes (Rakov and Uman, 2003). In case of +CG flashes the scenario is reversed, with negative charge ascending from the ground towards the cloud during the leader phase. A study by Le Vine and Krider (1977) showed that for -CG lightning flashes, the first RS exhibited robust radiation in the VHF range. The peak of VHF radiation at frequencies of 3, 139 and 295 MHz occurred at 10–30 µs after the initiation of the RS peak identified in the wideband (300 Hz – 2MHz) electric field waveform. They suggested that the VHF

radiation might be attributed to the effects of branches in the first stroke. Lapierre et al. (2014) addressed the issue of post-return stroke very high frequency radiation by examining the growth of positive leaders following the return stroke in -CG flashes, and explored its relationship with the continuing current (CC). Their research revealed that the positive leader propagates at a constant velocity before, during, and after the CC phase. However, they did not find a clear correlation between the positive leader growth and the presence of the CC. In a subsequent study, Lapierre et al. (2017) reanalyzed a seven stroke

-CG flash combining both Lightning Mapping Array (LMA) data and supplementary broadband interferometric data (INTF). The detailed mapping of the lightning flash highlighted the absence of negative in-cloud leaders following the RSs, while showing radiation from positive leaders, due to relatively good interferometer's ability to map the propagation of positive leaders. Their findings together with Stock et al. (2017) and Shao et al. (1995) show that immediately following a -CG return stroke, negative leader growth typically ceases. However, as the RS's potential wave reaches the top end of the lightning

channel in the cloud, it often triggers a burst of positive breakdown at the end of positive leader tips.

One of the sparse studies on electromagnetic radiation of +CG flashes is a study conducted by Shao et al. (1999). Their investigation primarily focused on the cloud-to-ground leaders preceding the RS, but they also showed the VHF activity following two +CG and -CG first return strokes. A robust radiation in the VHF power spectrum immediately after the first RS

in +CG flashes (in their Fig. 1), contrasting with the absent or minimal radiation in -CG flashes was recorded (in their Fig. 2), but this phenomenon was not described by the authors. A strong radiation following the RSs in +CG flashes was also shown in a work of Wu et al. (2018; 2022), who used a Fast Antenna Lightning Mapping Array (FALMA) and its 2D mapping results. A presence of numerous located sources during the time after the RS is shown in their studies for several flashes, with a suggestion of a fast negative leader propagating in the cloud being the source of this radiation. However, the information about

processes immediately following the RS (within the first 1 to 5 milliseconds) is missing because of the absence of 2D located sources during this temporal window. Tang et al. (2023) contributed to this topic by describing the post-RS behaviour of a +CG flash detected in China. They used interferometric measurements to study the spatial and temporal development of the flash, focusing on the detection of radiating VHF sources. After the RS, they identified two negative breakdown processes initiated at different branches of previously established but extinguished negative channels, supporting the bidirectional concept of lightning discharge. Another study investigating post-RS VHF radiation in a multi-stroke +CG flash is the work by Urbani et al. (2022). Following the RS and a subsequent RS, they observed a burst of VHF activity, which they suggested is associated with the CC phase. Urbani et al. (2022) also proposed that the RS brings ground potential to the channel, introducing the negative charge along the in-cloud leader channels. In their study mainly focused on the relationship between CC phase and in-cloud leader growth in +CG flashes, Lapierre et al. (2017) observed that in +CG flashes, the RS triggers an increase in the negative leader growth, both in number and speed. This phase is marked by strong VHF radiation from negative in-cloud leaders immediately following the +CG RS. The authors conclude that because negative leaders conduct significantly more current than positive leaders, the channel remains conductive during +CC phase, allowing the +CC flow to directly influence the growth rate of these negative in-cloud leaders. In contrast, for -CG flashes, the relatively low currents in positive leaders cause the channel to lose its conductivity very quickly (Lapierre et al., 2014, 2017; Stock et al., 2017), disconnecting it from the main channel, which is connected to the ground. As the in-cloud positive leader continues to grow, electric potential is built up until a K-event occurs, re-ionizing the channel. Despite these findings, there is still room for further exploration into understanding the post-RS dynamics of both +CG and -CG flashes.

The primary objective of this study is to gain complex understanding of electromagnetic activity occurring immediately after the first RS and to explore the distinctions between -CG and +CG lightning flashes. Integrating data from multiple sources allows us to obtain more extensive understanding of the processes occurring within the thundercloud after the first RS compared to previous works and to shed more light on properties of the less frequently occurring +CG flashes.

## 2 Instrumentation

We use broadband magnetic field measurements data obtained from the SLAVIA (Shielded Loop Antenna with a Versatile Integrated Amplifier) sensor, a magnetic loop, which detects the E-W horizontal component of the time derivative of the magnetic field fluctuations in a frequency range from 5 kHz to 37 MHz. The antenna has a cosine shaped sensitivity pattern and is connected to the radio receiver originally designed for the TARANIS spacecraft, sampling at a frequency of 80 MHz. The system operates in a triggered mode: 208-ms long waveform snapshots including a history of 52 ms are recorded when obtaining a trigger. The trigger is activated when the absolute value of the time derivative of the magnetic field exceeds a predefined threshold. Subsequently, these waveforms are numerically integrated, resulting in comprehensive representations of magnetic field evolution over time (Kolmasova et al., 2018; 2020). The precise timing of the sensor's clock is obtained via

a GPS receiver, accurate up to 1 µs. During the year 2015, the sensor was located at the Ersa station on the Corsica Island (altitude: 550 m, coordinates: 42.97°N, 9.38°E); the location is shown in Figure 1.

Narrowband VHF radiation emitted by lightning discharges is obtained from the lightning mapping array (LMA) network SAETTA (Suivi de l'Activité Electrique Tridimensionnelle Totale de l'Atmosphère) (Coquillat et al., 2019). This network consists of 12 stations strategically positioned across the Corsica Island (see station locations in Figure 1). Each station features an electric-field antenna, capable of detecting VHF radiation in the 60–66 MHz frequency range, sampled at 25 MHz. Individual stations store the time of arrival and power of the strongest VHF radiation peak detected within individual
consecutive 80-µs time intervals. Additionally, the total count of VHF sources surpassing a specified power threshold during each typical 80-µs interval, hereafter called VHF radiation rate, is recorded, with a maximum count of 2000 (i.e. sampling is performed with a cadence of 40 ns; sampling frequency is 25 MHz) (Rison et al., 1999; Kolmasova et al., 2018). The reception threshold is set for each station individually being determined by the local noise floor. SAETTA is capable of 3D localization of the VHF radiation sources, employing the time of arrival method from successful detections by at least six individual LMA
stations (Rison et al., 1999). As a result of this procedure, SAETTA provides 3D located sources, together with the estimations of their power. In this work, we use only sources obtained with a reduced chi-square value not greater than 1. Each LMA station is equipped with a GPS receiver that provides a time assignment accurate to 1 µs. The raw VHF signal recorded at each of the 12 LMA stations and the reconstructed VHF sources are used to document the studied flashes.

Supplementary data on 2D locations, polarities, and peak currents for analyzed CG discharges were supplied by the French lightning location system Météorage. This system using LS7002 sensors coupled to a total lightning processor (TLP) has a detection efficiency of 94% for CG lightning strokes, a median location accuracy of 120 m (Pedeboy et al., 2016), and an estimated accuracy of peak current amplitude around 18% (Schulz et al., 2016). In addition to CG discharges, Météorage detects the most powerful intracloud (IC) pulses corresponding to the vertical channel joining opposite charge areas in the
cloud. The mean detection efficiency for IC flashes (flashes where at least one cloud pulse is detected) is estimated to 56% (Pedeboy et al., 2018).

## 3 Methodology and data set

First, we conducted a visual examination of 1-ms plots of broadband magnetic field waveforms from 1728 triggered intervals,
each with a duration of 208 ms, recorded by the SLAVIA sensor from September to December 2015. This dataset contained 661 CG flashes. Then, for further analysis, we have selected all CG flashes with both distinguishable PB pulses (defined as a train of at least five bipolar pulses with peak-to-peak amplitude of at least 1 nT, appearing within 1 ms) and the first RS pulse in the SLAVIA waveforms that were simultaneously detected by the LMA and Météorage. The presence of the PB pulses is

required to ensure that the RS pulse is the first RS and not a subsequent RS pulse in a multi-stroke flash. This selection procedure resulted in 66 -CG and 26 +CG flashes.

## 4 Data analysis

In our investigation, we explored the relationship between broadband magnetic field waveforms and peaks of radiated VHF power detected by individual SAETTA stations. For all lightning flashes, we used their RS location by Météorage to correct the time of the measurements for the propagation delay between the signal detection by individual SAETTA stations and their detection by the SLAVIA sensor.

The map of all 66 -CG and 26 +CG analyzed flashes with their corresponding first RS polarities and RS peak currents $I_{RS}$ is shown in Figure 1a. Most of the flashes occurred in the vicinity of the SLAVIA sensor, in the northern part of the island, and above the sea or along the coastline (same as in Coquillat et al, 2019). This pattern is also influenced by an increased capability of the SLAVIA sensor to detect closer flashes and the topography of Corsica Island, where the presence of inland high mountains makes it more difficult for the SLAVIA sensor and some of the LMA stations to detect lightning occurring in the mountainous area or on the other side of the island. For -CG flashes the RS peak current was ranging from -6.3 kA to -260.0 kA, with the mean value of -63.9 kA and the median value of -54.3 kA. In case of +CG flashes, the RS peak current was ranging from 8.5 kA to 157.3 kA, with the mean value of 87.0 kA and the median value of 89.5 kA. Histograms of the first RS peak currents for all -CG and +CG flashes are plotted in Figure 1b and 1c, respectively. Furthermore, in all studied flashes, the initial polarity of PB pulses was the same as the polarity of the upcoming first RSs.

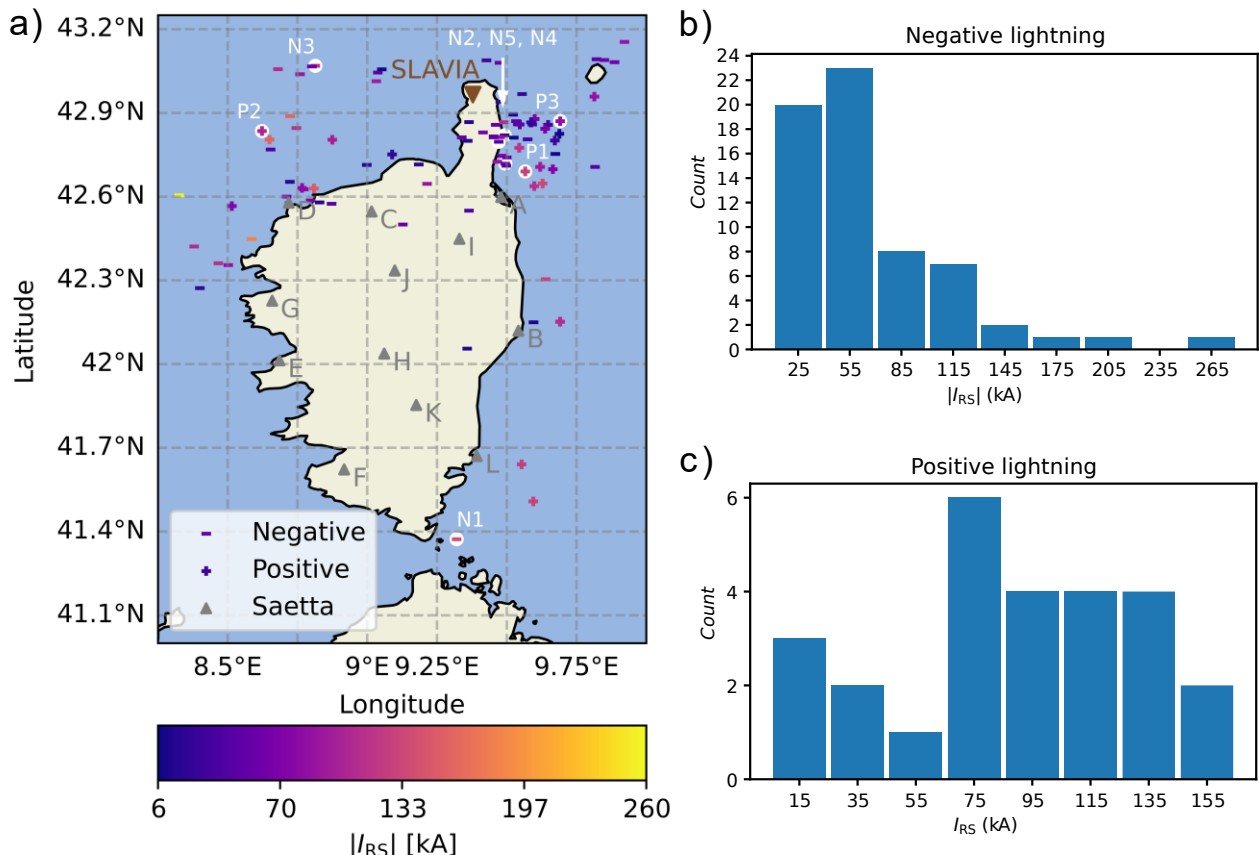

Figure 1: a) Map of all 1st return strokes used in this study as located by Météorage. -CG strokes are marked with -, +CG strokes with +. The colour corresponds to the absolute value of the first RS peak current $I_{RS}$ for each flash. The SAETTA stations are labelled by grey triangles with their corresponding letter name, while the SLAVIA sensor is marked as a brown reversed triangle. Cartographic outline denotes the island of Corsica and the northern part of Sardinia. b) Histogram of the absolute value of first RS peak current $|I_{RS}|$ for all 66 -CG flashes; c) histogram of $I_{RS}$ for all 26 +CG flashes.

## 4.1 Negative (-CG) flashes

### 4.1.1 Decrease of VHF radiation after RS

In the measurements of the majority (59 out of 66) of -CG flashes as recorded independently by separate LMA stations we observed a rapid decrease in the VHF radiation rate and power of VHF radiation sources following the first RS pulse peak. Specifically, both these quantities consistently reached a local peak within 2–30 µs after the wideband RS pulse peak, with the mean and median values of VHF power peaks recorded by the nearest LMA stations being -41.3 dBm and -40.7 dBm, respectively. Obtained mean and median VHF power peak values are similar to individual VHF power peak values detected during the initiation phase of -CG flashes or inverted IC flashes at the SAETTA stations closest to the discharge, as reported in Figs.3 and 4 in Kolmasova et al. (2018). Within a duration shorter than 300 µs, the counts decreased below 1500 VHF

samples above threshold (note that the LMA sampling cadence is 40 ns) detected in an 80-µs LMA interval. Unless there was another IC activity, such as a preparative process for a subsequent RS, the counts continued to decrease afterwards and in another 400 µs reached a low value ranging from 1 to 500 samples above threshold per 80-µs.

Examples of 3 -CG strokes detected by the SLAVIA sensor along with the sources of VHF radiation detected by one of the LMA stations closest to the RS strike point (as reported by Météorage) are plotted in Figure 2 as typical representatives of the observed behaviour. Each point marker symbolizes the strongest VHF source detected by a selected LMA station within an 80-µs LMA time interval. The colour of the marker corresponds to the VHF radiation rate detected within this specific time window. Each panel in Figure 2 represents the early stage of one selected -CG flash, along with information about its RS peak current and the location of the RS strike point, as reported by Météorage. Lightning N2 plotted in Figure 2b is a single stroke flash, lightning N1 (Figure 2a) and N3 (Figure 2b) are multi-stroke flashes with one subsequent RS and 2 subsequent RSs, respectively. The first RS strokes are highlighted in the Figure 1a and their distance from the SLAVIA sensor is indicated in the Figure 2 caption. The panels consistently show visible PB pulses followed by the RS pulse, as captured by the SLAVIA sensor (blue waveform). Across all three flashes shown in Figure 2 there is a significant sudden increase in VHF radiation rate and power of VHF radiation sources coinciding (within 250 µs) with the onset of the broadband PB pulses. This phenomenon serves as a reliable indicator of flash initiation (Kolmasova et al., 2018), as prior to the PB pulses, there is either an absence or a minimal presence of VHF radiation, typically perceived as environmental noise (Figure 2). At the time of the RS peak, both VHF radiation rate and power of detected VHF sources exhibit a peak followed by a rapid decrease. In these examples, the decrease is nearly immediate, with fewer than 500 40-ns detections within the 80-µs LMA time interval. This observed behaviour is consistent across 59 -CG flashes. Furthermore, it is noteworthy that in the time span from the lightning initiation to the occurrence of the first RS in flashes N2 and N3 (panels Figure 2b and Figure 2c), the VHF radiation rate reaches its maximum (2000 40-ns-counts) at the closest LMA station, meaning that every sample obtained with a cadence of 40 ns exceeded the threshold, revealing that there is almost continuous impulsive VHF radiation mainly from downward negative stepped leaders.

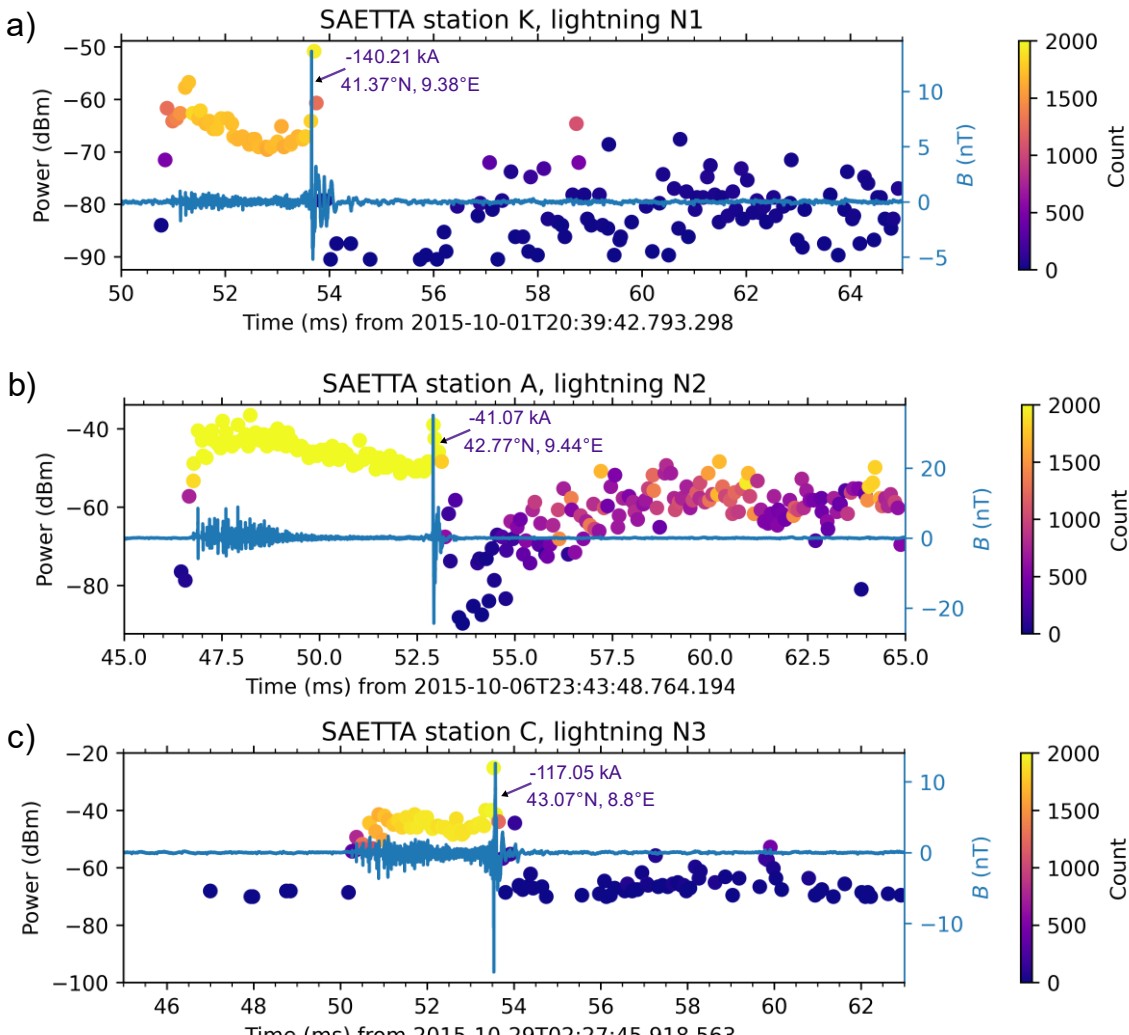

**Figure 2: Time evolution of the magnetic field as sensed by SLAVIA (blue waveforms) and VHF radiation amplitude measured at one LMA station around the time of the 1ˢᵗ RS of 3 different -CG flashes. Colors of the dots (dark blue – red – yellow colour scale) are proportional to the VHF radiation rate expressed in counts per 80-µs LMA time window. Geo-locations and peak current, as reported by Météorage, are indicated at the time of each RS. 1ˢᵗ RS distance from the SLAVIA sensor is for (a) lightning N1: 177.6 km; (b) lightning N2: 21.9 km; (c) lightning N3: 48.7 km.**

In flashes containing multiple RSs, we observed significant increase in both power and count of VHF radiation preceding these subsequent RSs, lasting for durations ranging from a few milliseconds to several tens of milliseconds. Figure 3 shows a typical example of a -CG flash with several RSs clearly captured by both SLAVIA and LMA. Figure 3a shows the detail of PB pulses and the first RS. Analogous to the examples portrayed in Figure 2, there is a sudden increase in both the counts and power of VHF radiation sources coinciding with the initiation of PB pulses. Furthermore, a local peak in the power of detected VHF sources is evident at the time of the first RS, followed by a rapid decrease of the VHF radiation rate to below 500 sources.

Figure 3b illustrates the entire flash with two additional subsequent RSs (no other RS was visible in the 208 ms long SLAVIA waveform, neither detected by Météorage). Notably, there is a significant increase in the VHF radiation rate and power of VHF sources detected by the closest LMA station A approximately 15 milliseconds prior to the magnetic field pulse emitted by the first subsequent negative RS (of -22.4 kA). During this interval, the number of counts within each 80-μs time interval reaches nearly 2000, a trend sustained until the occurrence of the subsequent RS pulse peak. At the time of the first subsequent RS, a

peak in VHF source power is noticeable, again, followed by a rapid decrease in both VHF radiation rate and power. Later on, approximately 28 milliseconds prior to the second subsequent RS pulse, another increase in VHF radiation rate and power is observed, persisting until the occurrence of the second subsequent RS. After the second subsequent RS, there is once again a decline in VHF radiation as recorded by the closest LMA station.

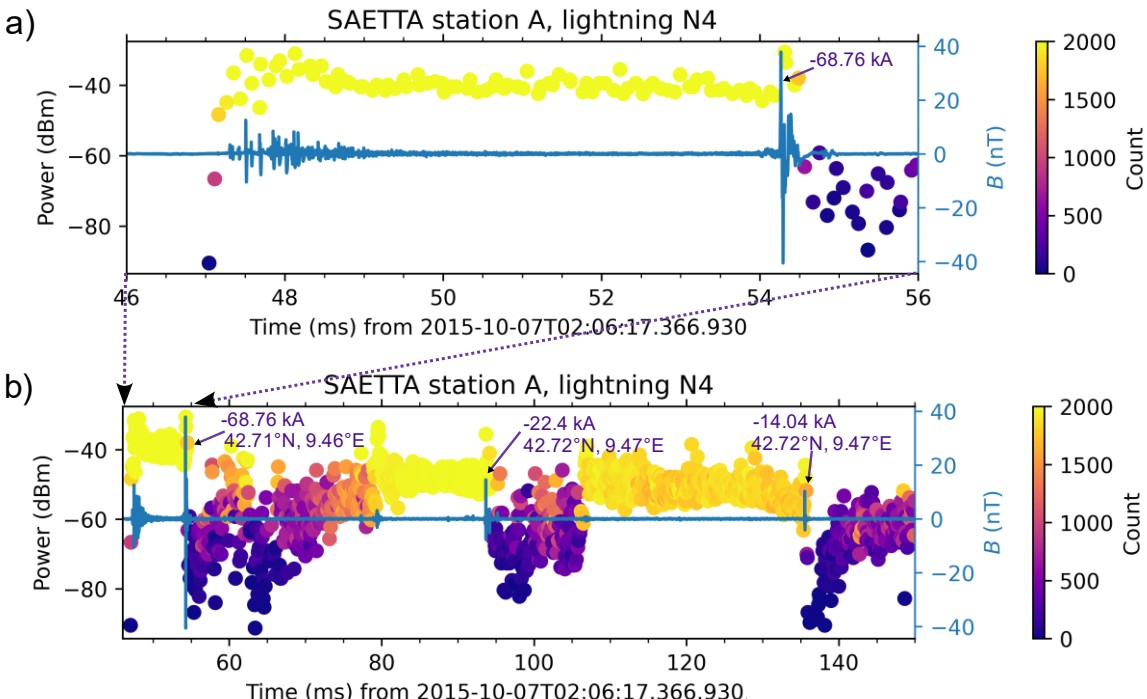

**Figure 3: Same as Figure 2 but for multi-stroke -CG flash as shown during (a) the early stage of the flash, with the PB pulses and the 1st RS and (b) the whole flash as detected by SLAVIA and Météorage. Distance from the SLAVIA sensor is for (a) 1st RS: 28.9 km; (b) 2nd RS (1st subsequent): 28.3 km; (c) 3rd RS (2nd subsequent): 28.3 km.**

### 4.1.2    Anomalous cases with strong VHF radiation after RS

In our observations, a deviation from the typical VHF radiation pattern was identified in 7 -CG flashes. In these cases, intense

VHF radiation lasted for more than 2 milliseconds after the wideband RS pulse peak. Among these 7 -CG flashes, in six cases, the amplitude of the largest PB pulse peak exceeded the amplitude of the RS pulse peak – indicating a strong lower positive charge center in the thundercloud, potentially weakening the RS (Nag and Rakov, 2009; Iudin et al., 2017; Tan et al., 2018;

Kolmasova et al., 2020). In addition, in three of these six cases, noticeable IC pulses were observed in the SLAVIA waveforms immediately following the RS pulse peak. In one of these cases, a relatively strong pulse appeared among these IC pulses. This

pulse was according to Météorage emitted by a negative RS stroke, nevertheless the pulse did not exhibit a typical RS shape and we consider it to be a pulse belonging to an IC discharge. In the other three cases, increased electromagnetic activity was evident in the raw SLAVIA data, signifying IC activity at higher frequencies. In the remaining one anomalous -CG flash, where the amplitude of the largest PB pulse peak did not exceed that of the RS pulse peak, increased electromagnetic activity was observed in the raw SLAVIA data.


As an example, Figure 4 shows a -CG flash exhibiting VHF radiation characteristics divergent from the typical pattern observed in the 59 -CG flashes analyzed in this study. In Figure 4a, the SLAVIA magnetic field waveform is depicted alongside the VHF radiation sources detected by the LMA station closest to the RS strike point, station A. The peak amplitude of the largest PB pulse peak exceeds that of the RS pulse. Consistent with the observed pattern in other -CG flashes, an increase in both

VHF radiation rate and power of VHF radiation sources is observed concurrently with the onset of PB pulses. While a local peak in VHF source power is observed at the time of the RS pulse peak, there is no significant decrease in either VHF radiation rate or VHF source power after the RS. The increased radiation persists until the end of the SLAVIA waveform capture (for over than 150 milliseconds after the RS pulse), accompanied by visible IC activity in the SLAVIA waveform, seen as sequence of irregular pulses. Notably, no subsequent RSs were identified in the SLAVIA data nor reported by Météorage for this flash.

Figure 4b presents the flash alongside all 3D located VHF sources provided by the LMA (depicted as colour marks) during the same 35-ms period. Despite continuous impulsive VHF radiation detected at LMA station A and VHF radiation detected at five other stations (though not continuous) during the 25-millisecond post-RS interval, only six VHF sources were reconstructed in 3D, and no 3D located sources were reconstructed prior to the RS.

The absence of 3D located sources during periods of high VHF radiation rates detected by individual LMA stations is a common characteristic observed across all analyzed flashes. Due to the lack of 3D located sources, precise tracking of the spatial development of the flash becomes nearly impossible, thereby preventing the accurate determination of the locations of IC channels emitting radiation in the time after the RS, similarly to results of Kolmasova et al. (2018) in their study of pre-stroke processes.


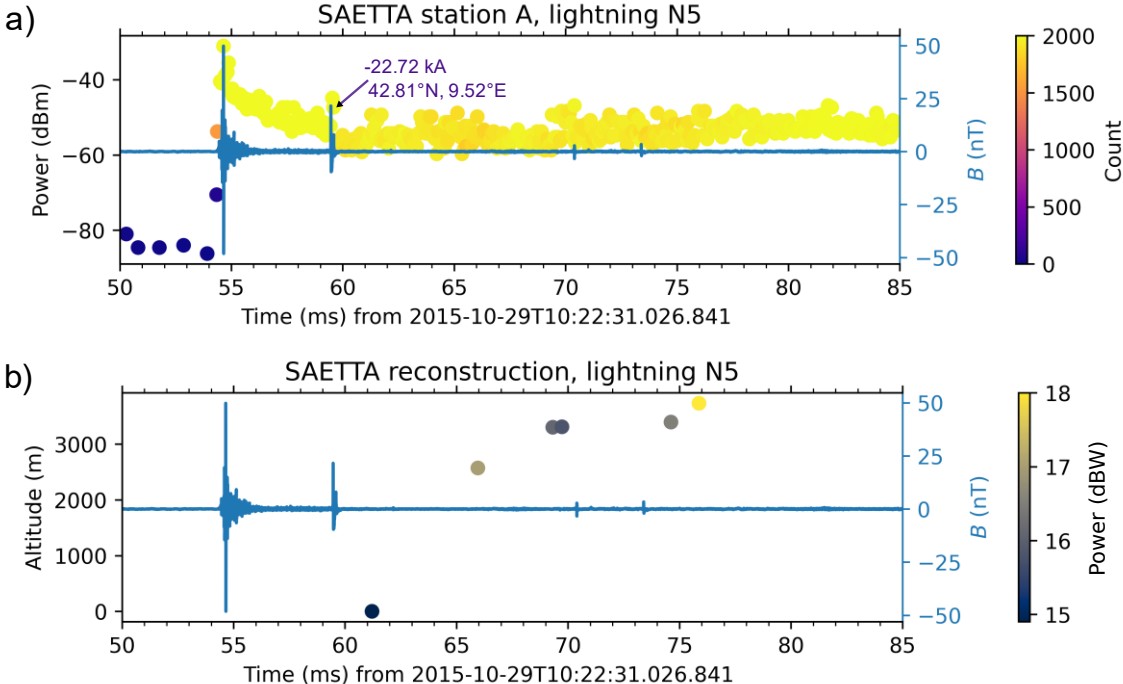

**Figure 4: (a) Same as Figure 2; (b) time-altitude of the VHF sources reconstructed during the same 35-ms period. Colors of the dots (dark blue – red – yellow colour scale) are proportional to the VHF radiation power measured per 80-µs LMA time window. 1st RS distance from the SLAVIA sensor is 20.9 km.**

**4.2 Positive (+CG) flashes**

In the LMA records of all 26 analyzed +CG flashes, an increase of VHF radiation was detected following the RS pulse, as illustrated in the selected examples shown in Figure 5. Specifically, the VHF power consistently reached its local peak within 1–34 µs after the wideband RS pulse peak. The mean and median value of VHF power peaks recorded by nearest LMA stations was -32.6 dBm and -31.6 dBm, respectively. Subsequently, strong VHF radiation in both counts and power persisted for tens to hundreds of milliseconds and was then followed by a slow decrease. The occurrence of strong VHF radiation implies a rapid stepwise expansion of in-cloud lightning channels in the aftermath of the first RS pulse.

As an example, and similarly to Figure 2, each panel of Figure 5 shows one single-stroke +CG flash with its RS peak current and location of RS strike point. All panels show visible PB pulses followed by the RS pulse, as detected by the SLAVIA sensor (blue waveform). Analogous to the -CG flashes illustrated in Figures 2, 3, and 4, each +CG flash in Figure 5 exhibits an abrupt increase in both the counts and power of VHF radiation sources coinciding with the beginning of PB pulses. This pattern is consistent across all studied -CG and +CG flashes, signifying the initiation of the flash, while negligible or weak VHF radiation, often identified as noise, is observed prior to the PB pulses. Additionally, a local peak in the power of detected VHF sources is observed at the time of the RS peak, a characteristic also shared by all analyzed -CG and +CG flashes.

Contrary to the behaviour observed in -CG flashes (see Figure 2), all +CG flashes depicted in Figure 5 exhibit robust VHF radiation (in counts and power) following the RS. For instance, in the +CG flash depicted in Figure 5a, the maximum count of 2000 VHF sources, only slowly decreasing to 1500 counts, endured for over 60 milliseconds after the RS. Similarly, in the flash depicted in Figure 5b, this phenomenon persisted for 10 milliseconds, while in the case of the flash shown in Figure 5c,

it endured for more than 30 milliseconds. This indicates nearly continuous impulsive VHF radiation following the RS. The slow decrease in power and counts of VHF radiation sources detected by individual LMA stations kept continuing afterwards.

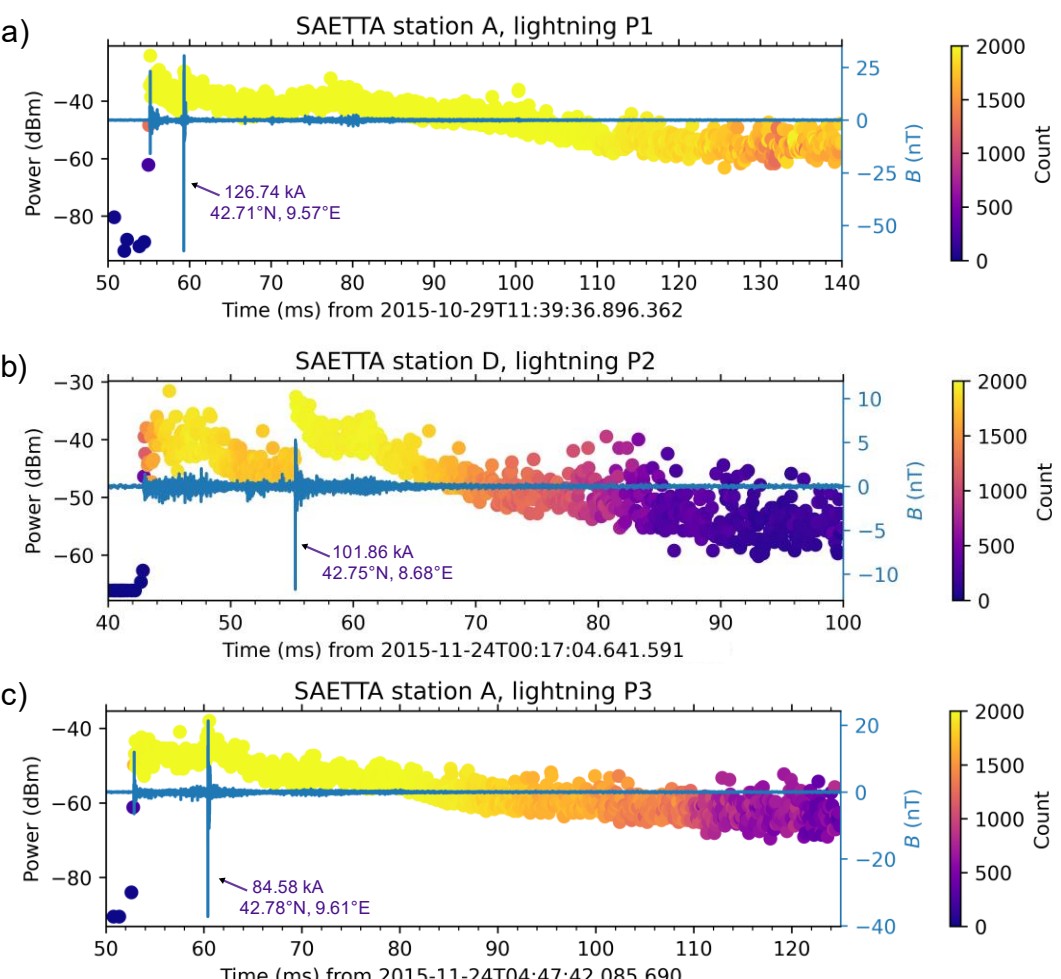

**Figure 5: Same as Figure 2 but for three different +CG flashes. RS distance from the SLAVIA sensor is for (a) lightning P1: 32.2**
**km; (b) lightning P2: 61.4 km; (c) lightning P3: 27.9 km.**

We were not able to track the detailed spatial propagation of the studied flashes, as SAETTA provides only a limited number of 3D located sources, especially during the stages of high VHF radiation. For the flash depicted in Figure 5a (flash P1), SAETTA did not provide any 3D located sources throughout the entirety of the high VHF radiation phase. However, for the flashes depicted in Figure 5b and 5c (P2 and P3), SAETTA successfully located 20 and 32 VHF sources (not shown in Figure 5), respectively, from the initiation of the flash until the end of the strong VHF radiation phase (when counts decreased to 1500 in the 80-µs LMA time interval).

In Figure 6, we present the 3D located VHF sources during the high VHF radiation phase for the flash P3, illustrated in Figure 5c. Figure 6a shows the SLAVIA magnetic field waveform together with the time evolution of the altitude of the reconstructed VHF sources, color-coded by time. Figures 6b and 6c show the spatial evolution of the flash together with the rainfall data. The 5-min rainfall data emanates from Météo-France radar 5-min Precipitation Amount composite (https://doi.org/10.25326/357). The RS was detected 180 seconds after the start of the 5-minute radar time window.

As seen in Figure 6, the VHF sources were reconstructed at rather low altitude between 2 and 3 km. An upward development is observed during the PB stage (designated as segment NL1), suggestive of an upward-moving negative leader. Afterwards, following the RS, multiple sources are detected at lower altitudes (ranging from 2 km to 2.5 km), with two distinct upward-moving groups of VHF sources (designated as segments NL2 and NL3), originating around the altitude of lightning initiation. Notably, as seen on the close-up map (Figure 6c), groups NL2 and NL3 are situated approximately 11 km away from the point of lightning initiation. Analysis of the 3D distribution of the VHF sources during the 4 minutes prior the studied flash, that include the studied flash as well, mainly revealed that lightning branches propagated fast horizontally at a height of 2.5 km within a rather extended 15 km x 15 km cloud region reaching about 11 km distance from the lightning initiation. Even though VHF radiation was detected by seven stations (five of them detected almost continuous impulsive VHF radiation), information regarding the flash's development after the RS is unavailable. These leaders probably emitted such strong VHF radiation, as evidenced in Figure 5c, potentially overwhelming SAETTA's 3D reconstruction capabilities. It is important to acknowledge that accurate tracking of the flash development using SAETTA data is hindered by the insufficient number of 3D located VHF sources. From Figures 6b and 6c, it is also evident that the flash occurred within one of the areas of higher accumulated precipitation, which is a typical observation for all the studied flashes.

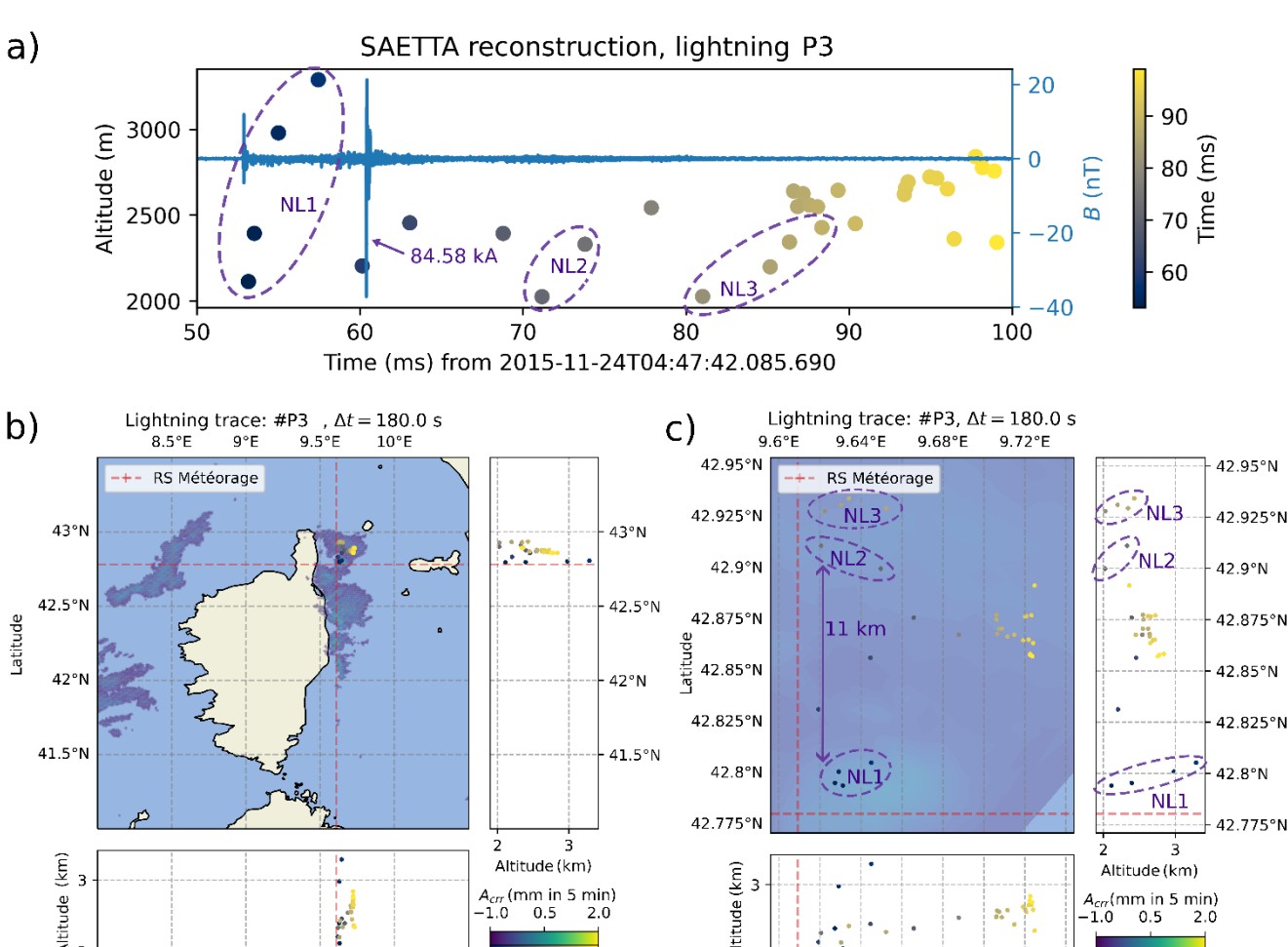

**Figure 6: +CG flash from Figure 5c (flash P3) detected by the SLAVIA sensor (blue waveforms) with the 3D located VHF radiation sources, provided by SAETTA (dots on the black – yellow colour scale): a) time-altitude development; b) map of Corsica island with the rainfall radar data, together with 3D located VHF sources and 2D location of RS strike point; c) close-up view on 3D located VHF sources with the rainfall data.**

## 5 Discussion and summary

We analyzed 26 +CG and 66 -CG flashes, focusing on the properties of VHF radiation just after their first RS. We found distinct behavioural patterns depending on the flash polarity: we observed a rapid decrease of the VHF source counts and VHF power immediately after the RS pulse peak in 59 (89 %) of -CG flashes, while in all inspected +CG flashes we observed strong VHF radiation lasting tens to hundreds of milliseconds after the wideband RS pulse peak.

Based on our experimental evidence and current understanding of characteristics of CG flashes, we can conclude as follows:

a. A strong VHF activity after the RS in +CG flashes shows that the negative component of the bidirectional leader is recharged and the negative in-cloud leaders propagate in stepped manner within the positive charge region. Our findings support the conclusion of Lapierre et al. (2017) that in +CG flashes, the RS triggers an immediate and significant growth of negative leaders in the cloud. This behaviour is schematically depicted in Figure 7 (blue discharges, flash b).

b. For -CG flashes, the situation is reversed; the in-cloud part of the bidirectional leader that gets recharged is the positive leader (see Figure 7, red discharge, flash a). The in-cloud positive leaders propagate within the negative charge center, with smooth, continuous VHF emission from their leader tips, which is hardly mapped in detail by LMA (Lapierre et al., 2014), though at some cases can be detected by interferometers (Pu et al., 2021; Stock et al., 2023). However, as the positive leaders advance, they emit weak, but impulsive radiation known as "needle" emissions, which LMAs and interferometers can mostly
detect and map (Hare, 2019, 2021; Pu, 2019; Edens, 2012; Van der Velde, 2013). Unfortunately, because of SAETTA's limitation by the Corsica mountainous relief and availability of LMA stations for the analyzed days, we lack sufficient 3D data on in-cloud leader propagation. This prevents us from clearly distinguishing radiation from negative in-cloud leaders to needle activity from positive in-cloud leaders. Data from individual stations across the 59 -CG flashes in our study showed an absence of VHF radiation immediately following RSs. The absence began within 100–700 µs after the RS and then persisted for at
least another 500 µs, often extending to several tens of milliseconds. This further suggest a lack of nearby negative leaders and that even close sensors generally failed to record needle activity from positive leaders during this period.

    Anomalous cases, where strong VHF radiation persisted for more than 2 ms after the negative RS pulse peak, can largely be explained by unique conditions associated with these specific flashes. For cases where the amplitude of the biggest PB pulse peak exceeds that of the RS pulse peak, we suggest the presence of a strong lower positive change center in the
thundercloud, which partially neutralised and thus weakened the RS, leaving a part of the originally downward moving stepped leader to propagate above the lower positive charge center inside the thundercloud (Iudin et al., 2017; Kolmasova et al., 2020; 2022). In one case where the amplitude of the biggest PB pulse is not larger than the amplitude of the RS pulse and no subsequent RS was detected, we, however, observed an increased electromagnetic activity in the raw (non-integrated) SLAVIA data. We can tentatively explain the strong VHF radiation by a close IC activity, possibly unrelated to the flash, radiating at
higher frequencies than can be seen in the SLAVIA integrated waveforms. However, as no VHF sources were reconstructed in 3D during times of intense VHF radiation, it remains uncertain whether in some of these anomalous cases, the sustained VHF radiation could indicate significant needle activity from recharged in-cloud positive leaders. Additionally, using rainfall radar data, we attempted to determine whether the size of the rainfall area is related to the duration of the strong VHF radiation following the negative RSs. Our investigation aimed to identify if larger rainfall areas are associated with prolonged durations
of VHF radiation, potentially clarifying the occurrence of the 7 anomalous -CG cases displaying robust VHF radiation after the RS. However, our analysis did not reveal any significant connection between these two factors.

The behavior of positive and negative in-cloud leaders after -CG and +CG RSs differs significantly: while in -CGs, the RS potential wave reaching the positive leader tips leads to recharging the in-cloud positive leaders without forming new

conducting channels (Stock et al., 2017), the RS potential wave in +CGs reaching a negative leader tip tends to produce dramatic channel structuring (Lapierre et al., 2017). The smooth, continuous emission from positive leader tips, although present, is generally not mapped or sometimes even undetectable by the LMA due to its low VHF radiation intensity. The simplified model (Figure 7) illustrates the hypothesized post-RS behavior of +CG and -CG flashes, acknowledging that.the actual charge distribution is much more complex, and the number and positioning of charge centers can vary significantly

(Stolzenburg and Marshall, 2008).

A similar hypothesis of lightning propagation after the RS was previously indicated by Shao et al. (1999), but was not investigated in detail. Moreover, Wu et al. (2018; 2022) and Lapierre et al. (2017) suggested an identical physical mechanism behind +CG flashes studied therein, which aligns with both the hypothesis proposed by Shao et al. (1999) and our own findings.

Here, we complement these studies by analyzing post-RS VHF processes in a larger data set, with a clear distinction between oppositely charged flashes, using the LMA data from individual stations. Further data, particularly with precise 3D VHF source localization using interferometric data, as demonstrated by Lapierre et al. (2017) or Tang et al. (2023), will be essential to validate the proposed hypothesis. Furthermore, observations from diverse geographic regions with varying lightning/storm climatology and across different seasons are essential to determine whether this hypothesis applies universally to all +CG and

-CG flashes.

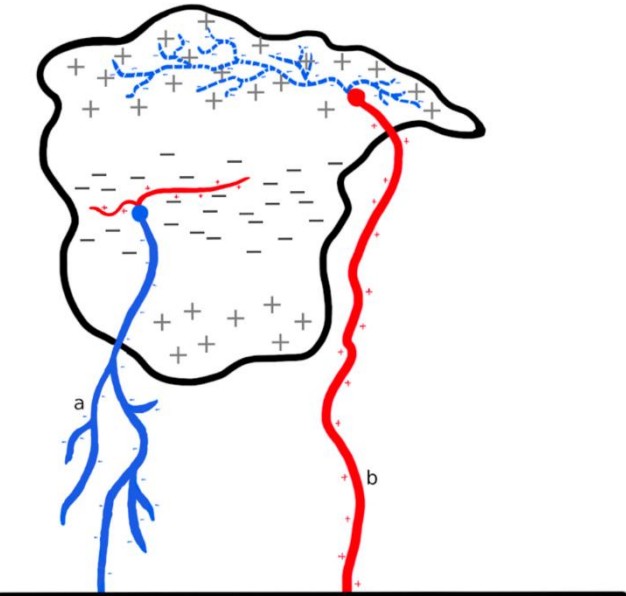

**Figure 7: Simple illustration scheme of lightning processes close to the first RS. a) negative RS with positive leaders propagating in cloud, b) positive RS with negative leaders in cloud.**

**Competing interests**

The authors declare that they have no conflict of interest.

**Author contribution**

IK and AK designed the study. AK, IK and ED interpreted the results. AK performed the data analysis and wrote the paper; all authors reviewed and edited the manuscript.

**Acknowledgements**

The work of AK was supported by GAUK grant GAUK198223. IK and OS acknowledge the support from the GACR grant 23-06430S. ED was supported by the CNES SOLID project.

This research was supported by the Johannes Amos Comenius Programme (OP JAC), project No. CZ.02.01.01/00/22_008/0004605, Natural and anthropogenic georisks.

AK thanks Mgr. Ales Podolnik, Ph.D. for useful discussions and assistance in data processing.

We would like to thank the reviewers, Michael Stock and Dylan Goldberg, for their valuable feedback and insightful comments, which have improved the quality of this manuscript.

Data availability:

SLAVIA and Météorage data, with SAETTA data from individual stations: doi: 10.17632/8dr67bw4mz.2

3D located SAETTA data - time and space locations of VHF sources (L1 data): https://doi.org/10.25326/236.

Météo-France radar database: https://doi.org/10.25326/357, maintained by the French national center for Atmospheric data and services AERIS.

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
