# Peer review of "Post-Return Stroke VHF Electromagnetic Activity in North-Western Mediterranean Cloud-to-Ground Lightning Flashes"

_EGUsphere, 2024_

## Author Comment (AC1)

**Answers to reviewer 1, Michael Stock (in red) with reviewer's comments (in black)**

Thank you very much for your thorough review and for the valuable suggestions. We have incorporated references to all the articles you mentioned and expanded both the introduction and discussion sections accordingly. The line numbers provided below refer to the revised manuscript with tracked changes.

**Specific Comments:**

Citations: This study omits discussion of the results in Lapierre et al 2017 (https://doi.org/10.1002/2016JD026189), and which deserves more than just a passing mention.

You have a citation from Lapierre's earlier paper on -CG leader growth, but skipped the second one looking at +CGs. As before, the paper is mostly interested in the behavior during continuing current, but for +CG flashes, this usually happens on the first (and only) stroke. This means the study has direct and detailed observations of exactly the behavior described in this paper. There is also a fairly detailed map of a -CG flash, extrapolated from LMA and INTF data. The INTF map lacks the initial return stroke, but the subsequent strokes all show the behavior discussed in this study.

Thank you very much for highlighting the study by Lapierre et al. (2017). In response, we have expanded both the introduction and discussion sections to incorporate their results, along with references to related studies by Stock et al. (2017), Shao et al. (1995), and others.

We expanded the introduction as follows:

- lines 49-56: we added "In a subsequent study, Lapierre et al. (2017) reanalyzed a seven stroke -CG flash combining both Lightning Mapping Array (LMA) data and supplementary broadband interferometric data (INTF). The detailed mapping of the lightning flash highlighted the absence of negative in-cloud leaders following the RSs, while showing radiation from positive leaders, due to relatively good interferometer's ability to map the propagation of positive leaders. Their findings together with Stock et al. (2017) and Shao et al. (1995) show that immediately following a -CG return stroke, negative leader growth typically ceases. However, as the RS's potential wave reaches the top end of the lightning channel in the cloud, it often triggers a burst of positive breakdown at the end of positive leader tips."
- lines 76-84: we also added "In their study mainly focused on the relationship between CC phase and in-cloud leader growth in +CG flashes, Lapierre et al. (2017) observed that in +CG flashes, the RS triggers an increase in the negative leader growth, both in number and speed. This phase is marked by strong VHF radiation from negative in-cloud leaders immediately following the +CG RS. The authors conclude that because negative leaders conduct significantly more current than positive leaders, the channel remains conductive during +CC phase, allowing the +CC flow to directly influence the growth rate of these negative in-cloud leaders. In contrast, for -CG flashes, the relatively low currents in positive leaders cause the channel to lose its conductivity very quickly (Lapierre et al., 2014, 2017; Stock et al., 2017), disconnecting it from the main channel, which is connected to the ground. As the in-cloud positive leader continues to grow, electric potential is built up until a K-event occurs, re-ionizing the channel."

We expanded and edited the discussion as follows:

negative leaders in the cloud."

- Lines 346-349: "As the presence of strong VHF activity indicates a stepwise propagation of negative in-cloud leaders, we speculate that after the RS in +CG flashes, the negative part of the bidirectional leader gets recharged. This leads to further propagation of in-cloud negative stepped leaders inside the positive charge center. ", changed to: "A strong VHF activity after the RS in +CG flashes shows that the negative component of the bidirectional leader is recharged and the negative in-cloud leaders propagate in stepped manner within the positive charge region. Our findings support the conclusion of Lapierre et al. (2017) that in +CG flashes, the RS triggers an immediate and significant growth of
- lines 353-364: "For -CG flashes, the situation is reversed, the cloud part of bidirectional leader that gets recharged is a positive leader (see Figure 7, blue discharge, a). The positive leader, propagating in the negative charge center is moving almost continuously and does not radiate in VHF. ",

changed to: "For -CG flashes, the situation is reversed; the in-cloud part of the bidirectional leader that gets recharged is the positive leader (see Figure 7, red discharge, flash a). The incloud positive leaders propagate within the negative charge center, with smooth, continuous VHF emission from their tips, which is hardly mapped in detail by LMA (Lapierre et al., 2014), though at some cases can be detected by interferometers (Pu et al., 2021; Stock et al., 2023). However, as the positive leaders advance, they emit weak, but impulsive radiation known as "needle" emissions, which LMAs and interferometers can mostly detect and map (Hare, 2019, 2021; Pu, 2019; Edens, 2012; Van der Velde, 2013). Unfortunately, because of SAETTA's limitation by the Corsica mountainous relief and availability of LMA stations for the analyzed days, we lack sufficient 3D data on in-cloud leader propagation. This prevents us from clearly distinguishing radiation from negative in-cloud leaders to needle activity from positive in-cloud leaders. Data from individual stations across the 59 -CG flashes in our study showed an absence of VHF radiation immediately following RSs. The absence began within 100–700 µs after the RS and then persisted for at least another 500  $\mu$ s, often extending to several tens of milliseconds. This further suggests a lack of nearby negative leaders and that even close sensors generally failed to record needle activity from positive leaders during this period."

- We have moved the paragraph from lines 305-320 in the previous manuscript version to lines 365-379 and edited it based on the studies you referenced and your suggestions. The main revisions are as follows: "As there are no 3D located VHF sources during the time of this flash, we are not able to assign the VHF radiation detected by individual LMA stations to the flash.", changed to: "However, as no VHF sources were reconstructed in 3D during times of intense VHF radiation, it remains uncertain whether in some of these anomalous cases, the sustained VHF radiation could indicate significant needle activity from recharged in-cloud positive leaders."
- lines 381-385: we added "The behavior of positive and negative in-cloud leaders after -CG and +CG RSs differs significantly: while in -CGs, the RS potential wave reaching the positive leader tips leads to recharging the in-cloud positive leaders without forming new conducting channels (Stock et al., 2017), the RS potential wave in +CGs reaching a negative leader tip tends to produce dramatic channel structuring (Lapierre et al., 2017). The smooth, continuous emission from positive leader tips, although present, is generally not mapped or sometimes even undetectable by the LMA due to its low VHF radiation intensity."

Expanded Discussion: The author's description of what they believe is happening in the cloud falls a little bit short of what I believe to be happening in the cloud.

The situation for +CGs is quite well documented in Lapierre 2017. Here the return stroke causes an immediate and significant growth or 'bloom' of negative leaders in the cloud. During the bloom, both the number of negative leaders and their speed increases. The growth period appears linked to the duration of the return stroke current, but even return strokes without continuing current will produce a bloom in the negative leader channels in the cloud.

For -CGs, there is no single reference to point to, but we can piece together the behavior from several studies. Immediately following a -CG return stroke, negative leader growth usually ceases. The potential wave from the return stroke itself traverses the entire lightning leader channel structure, all the way to the tips of the positive leaders. We know this become sometimes (perhaps even often) there will be a burst of positive breakdown at the positive leader tips immediately following the return stroke (Shao 1995 - although those observations are ambiguous and might be K-leaders, and Stock 2017 is not so ambiguous). In particular, Stock 2017 notes that the positive bursts following a negative return stroke do not appear to produce conducting channels.

From the Lapierre studies (Lapierre 2014, 2017 both), we know that after the return stroke the positive leaders continue to propagate with speed unaltered. Emission from the positive leader tips is usually not detectable (unless it is, Pu 2021, Stock 2023). Still, the LMA will detect emission on positive leaders in the form of needles, as they produce impulsive but weak RF (Hare 2019,2021, Pu 2019). Needle emission is detectable and mappable by the LMA, but most studies on this predate the term 'needles' (e.g. Edens 2012, Van der Valde 2013). In the case of SAETTA, the needle emission is weak enough that it's unlikely to be located by the network, but the nearest sensor should still be able to detect it in the logRF trace. Somewhat counterintuitively, renewing the channel conductivity of the positive leader channel with a stroke is usually associated with a strong reduction in needle activity. I don't think this has been directly shown following CGs (other than in this study), but it has been described after K-leader strokes which are similar (Jensen 2023). It then takes a while for needle activity to resume.

It is interesting that the negative and positive leaders behave so differently when subjected to very similar stimulus. Where the positive leader produces bright bursts of radiation, new conducting channels aren't usually formed. In contrast, the same potential wave arriving at the tip of a negative leader produces an immediate and dramatic effect which produces significant new channel structure.

Also Interesting, there is a similar effect to what the authors describe here seen in IC flashes after the negative leader stops propagating, even though there is no conductivity renewing stroke.

Shao 1995: https://doi.org/10.1029/94JD01943 Stock 2017: https://doi.org/10.1002/2016JD025909 Pu 2021: https://doi.org/10.1029/2021GL093145 Stock 2023: https://doi.org/10.3390/rs15143657 Hare 2019: https://doi.org/10.1038/s41586-019-1086-6 Hare 2021: https://doi.org/10.1029/2020JD034252 Pu 2019: https://doi.org/10.1029/2019GL085635 Edens 2012: https://doi.org/10.1029/2012GL053666 Van der Valde 2013: https://doi.org/10.1002/2013JD020257 Jensen 2023: https://doi.org/10.1029/2023JD039104 We have changed and expanded the discussion as mentioned above.

**Technical Comments:**

Line 97: Coquillat is a good citation for the SAETTA deployment, but the context indicates this to be a citation for how the LMA locates lightning (usually this is Rison 1999 Thomas 2004). You may want to revise this to be a little more clear.

Thank you, we have updated this reference, using Coquillat et al. specifically for the SAETTA deployment and adding citations to Rison et al. (1999) for the LMA lightning location methodology. (line 114).

Line 160: "Figure 1.a" should be "Figure 1a" to be consistent with other references

We have corrected this to "Figure 1a" (line 180) and ensured consistency across all other figure references in the manuscript.

Line 171: You discuss continuous emission in this paper in a few different contexts, some where the emission is continuous but impulse such that the LMA will well detect it, and in other places where the emission is smoothly continuous and the LMA may detect but not locate it. It may be good to be a little more careful distinguishing these modes.

Thank you for your suggestion. We have clarified our terminology to consistently specify when we are discussing continuous but impulsive radiation. Please see the revisions in lines 191, 241, 275, and 303.

Line 179: I think 'units' should be another word

We have replaced "units" with "a few milliseconds", (line 200).

Line 323: As noted above, I don't think you need much speculation about the presence of negative leaders.

We have revised this section: "As the presence of strong VHF activity indicates a stepwise propagation of negative in-cloud leaders, we speculate that after the RS in +CG flashes, the negative part of the bidirectional leader gets recharged..."

Changed to: "A strong VHF activity after the RS in +CG flashes shows that the negative component of the bidirectional leader is recharged and the negative in-cloud leaders propagate in stepped manner within the positive charge region.", (lines 346-347)

Line 328: The positive leader does produce impulsive radiation as it moves through the negative charge region in the form of needles (citations above). The tips also radiate, but usually aren't detectable (citations also above), this is a smoothly continuous emission and shouldn't be locatable by the LMA.

We have corrected and reformulated the passage for clarity: "The positive leader, propagating in the negative charge center is moving almost continuously and does not radiate in VHF."

Changed to: "The in-cloud positive leaders propagate within the negative charge center, with smooth, continuous VHF emission from their leader tips, which is hardly mapped in detail by LMA (Lapierre et al., 2014), though at some cases can be detected by interferometers (Pu et al., 2021; Stock et al., 2023). However, as the positive leaders advance, they emit weak, but impulsive radiation known as "needle" emissions, which LMAs and interferometers can mostly detect and map (Hare, 2019, 2021; Pu, 2019; Edens, 2012; Van der Velde, 2013).", (lines 354-358)

**Answers to reviewer 2, Dylan Goldberg (in red) with reviewer's comments (in black)**

**General Comments:**

This is an interesting study that analyzes VHF emissions immediately before and after the RS, and the results are important to the field. The introduction is comprehensive and clearly explains previous work and the primary objective of the study. The instrumentation section is concise and the data analysis is well-organized based on the discharge types and their differences (-CG, +CG). The data analysis and discussion sections could be strengthened further by adding a table of relevant statistics for VHF source counts, power, and durations.

Thank you for your thoughtful review and positive feedback on our manuscript. We appreciate your comments and suggestions, and we have addressed them below. The line numbers provided below refer to the revised manuscript with tracked changes.

Thank you for your suggestion regarding the addition of a table of relevant statistics for VHF source counts, power, and durations. However, because each SAETTA station exhibits different acquisition thresholds dependent on the level of the local electromagnetic noise, different distance between the SAETTA stations and the flashes, and the propagation of each signal is affected by the mountainous relief of Corsica Island, the detected VHF radiation is influenced by those environmental and range factors. This variability makes direct comparisons of VHF source counts, power, and durations across all SAETTA sensors problematic and unlikely to provide meaningful insights, except for pair or threesome of close LMA stations. Therefore, we do not include this statistic in our study.

**Specific Comments:**

It would be fascinating to see the analysis of this study with the addition of electric field data if it is available for the observed lightning discharges. It would also be interesting to see the mean or median VHF peak frequency of the strongest VHF source detected by the selected LMA station.

Thank you for your insightful suggestions. Unfortunately, electric field data are not available for the observed lightning discharges; we measured only magnetic field data.

Regarding the mean or median VHF peak frequency, it is not possible to get such information from the SAETTA system. LMA stations usually measure the total power of the signals within the 60-66 MHz frequency range. Therefore, analyzing the exact frequencies of the strongest sources is not possible.

If you were referring to the mean or median power of VHF radiation peaks observed at the times of RSs, we have added this information to the manuscript (see lines 165-168 and 258-259). However, as noted above, direct comparisons are problematic due to the variability caused by differences in acquisition thresholds across SAETTA stations, varying distances from the stations to the flashes, and the signal propagation through the mountainous terrain of Corsica. These factors significantly influence the detected VHF radiation, making such comparisons less meaningful.

**Technical Comments:**

Lines 135 - 140: Figure 1 resolution appears lower than other figures.

Thank you for pointing this out. We have increased the resolution of this figure. (line 153)

Line 179: "Units" should probably be changed to a different word.

We have replaced "units" with "a few milliseconds", (line 200).

Line 241: This should probably start as, "As an example," similarly to line 210.

Corrected, thank you, (line 263).

Lines 308 - 311: The beginning portion of this sentence could be worded to be more clear/concise.

We have revised the beginning of the sentence. The original phrasing has been changed to: "For cases where the amplitude of the biggest PB pulse peak exceeds that of the RS pulse peak, we suggest the presence of a strong lower positive change center...". (lines 366-368)

**Additional answers to both reviewers**

In addition to addressing the comments and suggestions of the reviewers, we made a few additional minor changes listed below to improve the manuscript. The most significant changes are listed below.

- Line 57: removed the sentence "In contrast, knowledge about +CG lightning flashes is significantly less comprehensive, lacking studies specifically focused on VHF radiation around the RS period (Nag and Rakov, 2012)"
- Lines 84-86: "Despite these findings, the authors did not investigate the post-RS behaviour more thoroughly, leaving room for further exploration into understanding the dynamics of +CG return strokes."

Changed to: "Despite these findings, there is still room for further exploration into understanding the post-RS dynamics of both +CG and -CG flashes"

- Line 251 (Figure 4): corrected mistake in units in Figure 4b, "dBm" -> "dBW"
- Line 385-389: "It is important to note that Figure 7 is a simplified illustration of our hypothesis regarding the behavior of +CG and -CG flashes after the RS. The actual charge distribution is

much more complex, and the number and positioning of charge centers can vary significantly (Stolzenburg and Marshall, 2008)."

Changed to: "The simplified model (Figure 7) illustrates the hypothesized post-RS behavior of +CG and -CG flashes, acknowledging that the actual charge distribution is much more complex, and the number and positioning of charge centers can vary significantly (Stolzenburg and Marshall, 2008)."

- Lines 395-398: "Further data is needed to confirm the outlined hypothesis of discharge propagation in the cloud shortly after the first RS. In particular, 3D localization of VHF sources is necessary for precise mapping of relevant lightning channels, for example, by using the data from interferometric detectors, as used by Tang et al. (2023). An extended data set recorded at different locations and in different conditions is needed to generalize our hypothesis." Changed to: ", using the LMA data from individual stations. Further data, particularly with precise 3D VHF source localization using interferometric data, as demonstrated by Lapierre et al. (2017) or Tang et al. (2023), will be essential to validate the proposed hypothesis. Furthermore, observations from diverse geographic regions with varying lightning/storm climatology and across different seasons are essential to determine whether this hypothesis applies universally to all +CG and -CG flashes."
- Line 512: added missing reference to Tan et al., 2018
- Line 523: doi added to reference to Wu et al., 2018